# Genomic Analysis of Response to Neoadjuvant Chemotherapy in Esophageal Adenocarcinoma

**DOI:** 10.3390/cancers13143394

**Published:** 2021-07-06

**Authors:** Fereshteh Izadi, Benjamin P. Sharpe, Stella P. Breininger, Maria Secrier, Jane Gibson, Robert C. Walker, Saqib Rahman, Ginny Devonshire, Megan A. Lloyd, Zoë S. Walters, Rebecca C. Fitzgerald, Matthew J. J. Rose-Zerilli, Tim J. Underwood

**Affiliations:** 1School of Cancer Sciences, Cancer Research UK Centre, Faculty of Medicine, University of Southampton, Southampton General Hospital, Southampton SO16 6YD, UK; fereshteh.izadi@swansea.ac.uk (F.I.); b.p.sharpe@soton.ac.uk (B.P.S.); s.p.breininger@soton.ac.uk (S.P.B.); j.gibson@soton.ac.uk (J.G.); r.c.walker@soton.ac.uk (R.C.W.); s.rahman@soton.ac.uk (S.R.); m.a.lloyd@soton.ac.uk (M.A.L.); z.s.walters@soton.ac.uk (Z.S.W.); M.J.Rose-Zerilli@soton.ac.uk (M.J.J.R.-Z.); 2Centre for NanoHealth, Swansea University Medical School, Singleton Campus, Swansea SA2 8PP, UK; 3Institute for Life Sciences, University of Southampton, Southampton SO17 1BJ, UK; 4UCL Genetics Institute, Division of Biosciences, University College London, Gower Street, London WC1E 6BT, UK; m.secrier@ucl.ac.uk; 5Cancer Research UK Cambridge Institute, University of Cambridge, Cambridge CB2 0RE, UK; ginny.devonshire@cruk.cam.ac.uk; 6MRC Cancer Unit, Hutchison/MRC Research Centre, University of Cambridge, Cambridge CB2 OXZ, UK; RCF29@MRC-CU.cam.ac.uk

**Keywords:** esophageal adenocarcinoma, chemotherapy, mutation, response, genomics, NAV3

## Abstract

**Simple Summary:**

Esophageal adenocarcinoma (EAC) is a cancer with very poor survival outcomes. Patients are treated with pre-operative chemotherapy or chemoradiotherapy before surgery. However, four out of every five patients do not respond to pre-operative therapy and these patients (non-responders) have significantly worse outcomes. Identifying non-responders prior to therapy would allow alternative treatment pathways to be offered to these patients. In this study, we analyze whole genome sequences of pre-treatment biopsies from 65 patients and find that non-responders display chromosomal instability and increased gene copy number alterations. We report a distinct profile of copy number alterations in non-responders compared to responders, predominantly in genes involved in cell cycle control and RTK/Ras signaling. Mutations in the tumor suppressor *NAV3* are also found exclusively in non-responders. These genetic profiles present potential drug targets for investigation in EAC patients who would not respond to pre-operative chemotherapy.

**Abstract:**

Neoadjuvant therapy followed by surgery is the standard of care for locally advanced esophageal adenocarcinoma (EAC). Unfortunately, response to neoadjuvant chemotherapy (NAC) is poor (20–37%), as is the overall survival benefit at five years (9%). The EAC genome is complex and heterogeneous between patients, and it is not yet understood whether specific mutational patterns may result in chemotherapy sensitivity or resistance. To identify associations between genomic events and response to NAC in EAC, a comparative genomic analysis was performed in 65 patients with extensive clinical and pathological annotation using whole-genome sequencing (WGS). We defined response using Mandard Tumor Regression Grade (TRG), with responders classified as TRG1–2 (*n* = 27) and non-responders classified as TRG4–5 (*n* =38). We report a higher non-synonymous mutation burden in responders (median 2.08/Mb vs. 1.70/Mb, *p* = 0.036) and elevated copy number variation in non-responders (282 vs. 136/patient, *p* < 0.001). We identified copy number variants unique to each group in our cohort, with cell cycle (*CDKN2A*, *CCND1*), c-Myc (*MYC*), RTK/PIK3 (*KRAS*, *EGFR*) and gastrointestinal differentiation (*GATA6*) pathway genes being specifically altered in non-responders. Of note, *NAV3* mutations were exclusively present in the non-responder group with a frequency of 22%. Thus, lower mutation burden, higher chromosomal instability and specific copy number alterations are associated with resistance to NAC.

## 1. Introduction

Esophageal adenocarcinoma (EAC) is a cancer of unmet clinical need. Patients with locally advanced EAC suitable for curative treatment receive neo-adjuvant chemoradiotherapy or neo-adjuvant chemotherapy (NAC) with or without adjuvant chemotherapy as standard of care. Randomized trials of NAC have consistently shown survival benefits for patients [1,2,3,4,5]. However, this survival advantage (9% at five years) [6] is not due to an incremental improvement in outcome for all patients, but instead driven by a very good response in less than 20% of patients treated with the ECX regimen [7,8,9,10,11], or in 37% of patients treated with the FLOT regimen [11]. Primary tumor regression following neoadjuvant therapy (NAT) can be measured using the Mandard Tumor Regression Grade (TRG) in resected specimens after surgery [12,13,14] and is informative for both disease-free and overall survival [13,14,15]. A Mandard TRG score of 1 corresponds to complete regression of the tumor leaving only fibrosis; TRG 2 is defined by fibrosis and scattered residual tumor cells; TRG 3 and 4 display progressively less tumor regression; and TRG 5 tumors display no regressive changes in response to NAT [12]. Genetic mechanisms associated with tumor response to NAT have been assessed in a variety of different cancer types [16,17,18,19,20,21,22] including rectal adenocarcinoma, but have not been widely investigated in EAC. Predictive biomarkers of response following NAT have been proposed for EAC, including functional imaging and expression of genes regulating apoptosis, angiogenesis, cell cycle, and DNA repair as well as growth factors and their receptors, but none have approached clinical practice [13,14].

EAC genomes are characterized by a high degree of chromosomal instability [23,24], and large-scale genomic studies, such as those conducted by the OCCAMs UK consortium using whole genome sequencing to contribute the International Cancer Genome effort, have identified key driver genes and clinically relevant biomarkers for prognostication [24,25]. This large cohort with extensive data on treatment and clinic-pathological response provides an ideal opportunity to investigate the predictors of response and resistance to chemotherapy. To date, only two studies using whole exome sequencing have investigated genetic features associated with response to NAC in which genetic bottlenecks, intratumor heterogeneity and early chromosomal instability were found to be related to NAC response/resistance in EAC [26,27]. These studies provide key insights into the genomic evolution of EAC through NAC and the changes in the genome architecture following clinical response. There is a need for studies to further characterize the whole genomic landscape in EAC at the time of diagnosis (pre-treatment) to enable identification of predictive biomarkers for response to NAC and to identify the consequences of genomic lesions suitable for novel interventions.

Here, we describe results from whole genome sequencing (WGS) of pre-treatment biopsies from 65 EAC patients treated with NAC and surgery, alongside RNA-seq, to investigate the genetic features associated with NAC response. We describe a hierarchical approach to the comparative analysis of the genomes of EAC responders and non-responders, starting with total mutational burden, continuing through large-scale chromosomal events and on to driver gene mutations, before defining the key genomic differences between groups and their potential for possible therapeutic intervention.

## 2. Materials and Methods

### 2.1. Overview of Patients and Sequencing Strategy

EAC patients in this study are presented in Figure 1A & Table 1. Sample collection and processing were performed as previously described [28] as part of the OCCAMS (Oesophageal Cancer Clinical and Molecular Stratification) Consortium. Pathological tumor response was assessed in the resection specimens by tumor regression grading (TRG) [12] with responders defined as TRG ≤ 2 and non-responders as TRG ≥ 4. Mandard grade was scored by a specialist gastrointestinal pathologist blinded to the clinical data at the treating cancer center.

### 2.2. Whole-Genome Sequencing Analysis

WGS, single nucleotide variant (SNV) and small insertion or deletion (Indel) calling was performed using Strelka [29] (version 2.0.13; Illumina, San Diego, USA, 2012.) against the GRCh37 reference genome as described by Secrier et al. [24], with 94% of the known genome being sequenced while achieving a PHRED quality of at least 30 for at least 80% of mapping bases.

Functional annotation of the resulting variants was performed using ANNOVAR [30] and the Ensembl Variant Effect Predictor (VEP) (https://www.ensembl.org/Tools/VEP, release 75; date accessed: 1 November 2020). Furthermore, 536 false positive genes [31] were removed from subsequent analysis. Data visualization including oncoprints and lollipop plots were performed by maftools (version 2.4.15; Singapore, 2018.) [32]. Mutually exclusiveness or co-occurrence analysis of genes was also performed by maftools using pair-wise Fisher’s exact test to detect significant pairs of given genes.

### 2.3. Copy Number and Clonality Analysis

Absolute genome copy number following correction for estimated normal-cell contamination was called using ASCAT package (version 2.5.2; Oslo, Norway, 2010.) in R [33]. Cellularity and ploidy estimates were also obtained using ASCAT and samples with estimated cellularity <20% were removed from further analysis. Significantly amplified/deleted regions in the cohort were identified using GISTIC2.0 (version 2.0; Cambridge, MA, USA, 2011.) [34]. Copy number variations (CNVs) were corrected for ploidy (= total copy number of the segment/average estimated ploidy of each sample) and GISTIC was run on an input defined as the log2 of the CNV with gain (-ta) ≥ 1.0 and loss (-td) ≤ 0.4, respectively.

### 2.4. Genomic Instability Analysis

Copy number burden represents the fraction of bases deviating from baseline ploidy (defined as above 0.5 or below –0.5 in log2 relative copy number space and in segments >1 kb length) named as genomic instability score (GII). CIN70 score was calculated by averaging the FPKM expression values of CIN70 signature in each sample from available RNA-seq data.

### 2.5. Mutational Signature and Neoantigen Analysis

Tumor Mutation Burden (TMB) in terms of per megabases was measured with 50 MB capture size for non-synonymous mutations. To compare the neoantigen load between the two groups, we used binding affinity for patient-specific class I human lymphocyte antigen (HLA) alleles, constituting potential candidate neoantigens by checking for the binding strength for peptides of length 9 using NeoPredPipe (version 1.1; Tampa, FL, USA, 2019) [35]. We then quantified the peptides that displayed high affinity (recognition potential > 1) binding in tumor, but no binding in the respective matched normal [36] as recognition potential prediction step implemented in NeoPredPipe to obtain recognition potential for each sample. We then compared the recognition potential between two groups by Wilcoxon rank-sum test. EAC mutational signatures were extracted using SigProfiler (version 1.1.0; San Diego, CA, USA, 2020) [35]. We processed the COSMIC solution to remove any artefactual signatures and signatures that contribute on average less than 1% of the mutations in the genome. A total of nine mutational signatures were identified, of which six were previously identified in EAC described by Secrier et al. [24]: SBS17A and SBS17B dominated by T > G substitutions in a CTT context and possibly associated with gastric acid reflux; SBS3, a complex pattern caused by defects in the *BRCA1/2*-led homologous recombination pathway; SBS2, C > T mutations in a TCA/TCT context, an APOBEC-driven hypermutated phenotype; SBS1, C > T in a *CG context, associated with aging processes; an SBS18-like signature, C > A/T dominant in a GCA/TCT context, formerly described in neuroblastoma, breast and stomach cancers; SBS13 is usually found in the same samples as SBS2; SBS5, linked to tobacco exposure; and SBS41 with unknown etiology. Clustering of mutational signatures was performed with the NMF package (version 0.23.0 Cape Town, SA, USA, 2010) [37] set to three main clusters as previously observed in EAC.

### 2.6. Expression Profiling by Bulk RNA Sequencing (RNA-seq) and Gene Set Enrichment Analysis (GSEA)

We were able to explore gene expression changes to investigate the expression levels of recurrently amplified/deleted genes, between responders and non-responders in 30 RNA-seq samples matched with WGS data (responders/9, non-responders/21). For a given EAC known driver identified as recurrently amplified or deleted in either group, we compared fragments per kilobase of transcript, per million mapped reads (FPKM) values for that gene by Wilcoxon rank-sum test. For GSEA by using MSigDB hallmark gene sets, we used normalized values from DESeq2 as input. We used GSEA and DESeq2 modules both implemented online (https://www.gsea-msigdb.org/gsea/; date accessed: 1 December 2020) [38] with default parameters.

### 2.7. Survival Analysis

For relating CNVs to overall survival, we used a Boolean matrix of CNV status of EAC driver genes. Multivariate analyses were performed by Cox proportional hazards regression model using the survival package (version 3.2.7) in R. Overall survival was defined as the time interval from initial surgical excision to death or last follow-up time (censored) and Kaplan–Meier plots were visualized using the ggkm 1.0 R package (https://github.com/michaelway/ggkm; date accessed: 1 December 2020).

### 2.8. DDR Pathway Deregulation Analysis

The Pathifier algorithm (version v1.0; Rehovot, Israel, 2013.) calculates for any given pathway a deregulation score (PDS) for each cancer sample, based on gene expression data (log2 normalized) [39]. Only the 5000 genes with the largest variation over available RNA-seq samples were used as input to the algorithm. PDS score represents the extent to which the activity of the pathway differs in a particular sample from the activity in the opponent sample. Here responders and non-responders were used as opponent groups of samples. We calculated an average of PDS over 16 DDR sub-pathways by using more than 450 genes associated with DDR, as previously described in a pan-cancer analysis [40].

### 2.9. Classification of Genes Relevant for Genomics-Driven Therapy

To identify genes relevant for genomics-driven therapy, we used version 2.0 of TARGET (tumor alterations relevant for genomics-driver therapy) database (www.broadinstitute.org/cancer/cga/target). We also used OncoKB [41] (https://www.oncokb.org/; date accessed: 1 January 2021.) for the association of drug-biomarkers of differentially mutated genes.

### 2.10. Identification of Specific Mutations with Therapeutic Relevance

The DoCM [42] was used to identify mutations with clinical evidence (drug targets associated with a mutation; diagnostic or prognostic markers associated with a mutation) or functional evidence (disease function described in cell lines; disease function described in animal models). The database is available online at docm.genome.wustl.edu (date accessed: 1 November 2020).

### 2.11. Statistics

Measurements between groups were compared using the Wilcoxon rank-sum test for continuous data with non-normal distribution and T-test for data with normal distribution or Fisher’s exact test for count data.

## 3. Results

### 3.1. Patient Characteristics and Overall Study Design

We selected 65 cases from the well-curated OCCAMS consortium multi-center dataset [24,25] that had available Mandard Tumour Regression Grading (TRG) information and WGS data from pre-treatment biopsies. These were classified into two groups based on TRG: 27 responders (TRG1 (*n* = 18) and TRG2 (*n* = 9)) and 38 non-responders (TRG4 (*n* = 28) and TRG5 (*n* = 10)) (Figure 1A). We excluded TRG3 classified cases because of their prognostically heterogeneous behavior [43]. A summary of the clinicopathological data for the cohort is shown in Table 1 with full details available in Appendix A. Median follow-up in the cohort was 56.7 months (1.5–78.8 months). In line with our previous multi-center cohort study [7], TRG defined responders had favorable prognosis compared to non-responders with a significantly longer overall survival (78.5 vs. 33.8 months, *p* < 0.001, Figure 1B). As expected, following NAC, non-responders had higher pathological TNM stage (ypT and ypN) compared to responders (χ^2^ test, *p* = 0.001, Table 1) but there was no difference in pre-NAC TNM stage [44]. All 65 patients had WGS data generated from endoscopic biopsies and matched germline DNA taken at the time of cancer diagnosis and before any treatment. In total, 9 responders and 21 non-responders had matched RNA-seq data to complement the WGS dataset according to the availability of tissue.

### 3.2. Mutational Profiles Associated with Response to NAC

To establish an overview of the mutational landscape associated with response to NAC, we compared the genomes of responders and non-responders. We performed assessment of somatic copy number variations (CNVs), small insertions/deletions (indels) and variant calling of somatic single nucleotide variants (SNVs) as previously described [24,25,28,45]. Two non-responders were found to have microsatellite instability (MSI) (Patient 27, score = 5.63 and Patient 29, score = 3.44) and these were excluded to avoid potentially confounding effects on statistical analyses, which could compromise assessment of the other tumors [24,46]. MSI-high tumors are known to respond to immune checkpoint blockade, providing a potential treatment pathway for these patients [47].

Initially, we investigated overall mutation burden in responders and non-responders on a genome-wide level. We identified a median of 104 (3–286) non-synonymous mutations per tumor genome in responders, compared to 85 (1–171) mutations in non-responders and the mutation frequency per megabase (Mb) was higher in the responder group (2.08 (range: 0.14–3.66) vs. 1.7 (range: 0.02–3.42); Figure 2A, Wilcoxon rank-sum test, *p* = 0.036).

To investigate the mutational profile of SNVs in the trinucleotide context in our cohort, we performed mutational signature extraction using SigProfiler (version 1.1.0; San Diego, CA, USA, 2020.) [35]. Nine mutational signatures were defined in our cohort (Figure 2B). We hierarchically clustered our cases, and in agreement with prior studies [24], three main subgroups were observed corresponding to predominant signatures: these are classified as C > A/T dominant (SBS1/5 and SBS18); DDR impaired (SBS3); and mutagenic (SBS17A/B) subgroups. There was no significant difference in the proportion of responders and non-responders between subgroups (Chi-squared test, *p* = 0.4). However, the DDR impaired subgroup is defined by signature 3 mutations, and the majority of these tumors (3/4) were non-responders. Signature 3 is associated with failure of DNA double-strand break-repair by homologous recombination, which could lead to chromosomal instabilities. Consequently, we assessed the dysregulation of DDR pathways using gene expression data of available RNA-seq samples (*n* = 9 responders, *n* = 21 non-responders) and created a pathways dysregulation score (PDS) using Pathifier (version v1.0; Rehovot, Israel, 2013) [39]. Non-responders exhibited greater dysregulation in DDR pathways compared to responders (Figure 2C, Wilcoxon rank-sum test, *p* = 0.002).

Having observed higher mutational burden in responders, we hypothesized that this would also be correlated with a greater neoantigen load. We used NeoPredPipe (version 1.1; Tampa, FL, USA, 2019) [48], a predictive tool, to identify tumor neoantigens using binding affinity for patient-specific class I human lymphocyte antigen (HLA) alleles. We next calculated the neoantigen recognition potential by quantifying the peptides that displayed high affinity binding in tumors but had no predicted binding in the matched normal sample [36]. Considering only those samples with recognition potential value above 1, responders had a significantly higher neoantigen recognition potential score (Wilcoxon rank-sum test, *p* < 0.001, Figure 2D), possibly supporting previous associations between CD8+ tumor infiltrating lymphocyte levels and improved survival in EAC following NAC [24,49].

#### 3.2.1. Non-Responders Have More Chromosomal Instability and Unique Copy Number Alterations

We next moved to consider chromosomal and copy number events and correlate these with mRNA expression, where possible, before considering point mutations. To investigate correlations between chromosomal instability (CIN) and response to NAC, we measured the proportion of the genome affected by copy number alteration. Non-responders exhibited a higher level of CIN as evidenced by a higher Genomic Instability Index (GII) [50] (Wilcoxon rank-sum test, *p* < 0.001, Figure 3A). To confirm these findings, we evaluated the CIN70 signature in matched RNA-Seq data, a gene signature whose expression was consistently correlated with total functional aneuploidy across multiple cancer types [51]. We found a higher CIN70 signature in non-responders, but this was not significant (*p* = 0.064), likely due to the small size of the RNA-Seq cohort.

We then identified recurrently amplified or deleted regions using GISTIC2.0 [34]. In responders, a total of 3626 CNVs were detected (median 136/patient, range: 0–292) including 2961 amplifications (median 115/patient, range: 0–286) and 665 deletions per case (median 28/patient, range: 5–53). In non-responders, there were a total of 9637 CNVs (median 282/patient, range: 5–504) including 6198 amplifications (median 185.5/patient, range: 0–382) and 3439 deletions (median 99.5/patient, range: 0–239). The total CNVs and the number of amplifications and deletions were higher in non-responders (Wilcoxon rank-sum test, *p*-values < 0.001, 0.025 and 0.001 respectively, Figure 3B). At the chromosomal arm level, we found recurrent amplifications of chr 20q in 48% of responders (FDR < 0.1), while we identified no significant amplification in non-responders (Appendix A). Furthermore, we found 13 and 20 significant deletions of chromosomal arms in responders and non-responders, respectively. Deletion of 14p was unique to responders, and deletions of 8p, 9q, 9p, 10q, 15q, 16p, 19q, and 22q were unique to this cohort of non-responders, showing a higher level of large-scale deletion in non-responders.

We next looked at copy number alterations in 76 previously validated EAC driver genes [25]. We restricted our initial analysis to these 76 genes as they have been comprehensively analyzed and verified in contemporaneous and clinically relevant cohorts in addition to downstream functional biological assessment. We identified significantly amplified or deleted peaks for the two groups (FDR < 0.1, Figure 3C,D, Appendix A). Distinct focal amplifications and deletions in EAC driver genes are illustrated in Figure 3D. The responders contained two unique amplification peaks: 17q12, containing *ERBB2* (FDR < 0.001) and 19q12, containing *CCNE1* and *TSHZ3* (FDR = 0.003) (Appendix A). These focal amplifications contrast with observed chromosome arm deletions in 19q observed in non-responders. Meanwhile, the non-responders contained more unique amplification peaks: 18q11.2 containing *GATA6* (FDR < 0.001); 7p11.2 containing *EGFR* (FDR = 0.018); 11q13.3.2 containing *CCND1* (FDR < 0.001); 12p12.1 containing *KRAS* (FDR < 0.001); 6q23.3 containing *MYB* (FDR  =  0.085); and 8q24.21 containing *MYC* (FDR  =  0.007) (Appendix A). Focal amplifications in MYC and GATA6 in non-responders contrast with our findings of arm level deletions in responders at the same chromosome arm, 18q. We investigated whether copy number changes in driver genes were co-occurrent and found that *GATA6* and *EGFR* were co-occurrent in non-responders (Fisher’s exact test, *p* = 0.002, Appendix A).

#### 3.2.2. mRNA Expression Level Supports the Dysregulation of EAC Driver Genes in Non-Responders

We reasoned that if these amplification/deletion peaks played a role in affecting the response to NAC, then we would observe corresponding signals in their related downstream pathways and patient survival would be affected. To do this we used matched RNA-seq data (*n* = 9 responders and *n* = 21 non-responders) and compared the FPKM values (fragments per kilobase of transcript per million mapped reads) of recurrently amplified and deleted EAC driver genes between groups. Consistent with copy number alterations, we observed the upregulation of CDK6 (Wilcoxon rank-sum test, *p* = 0.004), CCND1 (*p* = 0.004), GATA4 (*p* = 0.037) and MYC (*p* < 0.001) at the transcript level in non-responders (Figure 4A). Furthermore, patients with amplification at the corresponding chromosomal regions of cell cycle regulators had a worse prognosis with a shorter overall survival, including CCND1 (median survival 20.8 months in CCND1 amplified samples vs. 78.5 months in CCND1 neutral samples, *p* = 0.007), CDK6 (median survival 33.8 months in CDK6 amplified samples vs. 73.0 months in CDK6 neutral samples, *p* = 0.01) and deletion of regions harboring CDKN2A (median survival 33.8 months in CDKN2A deleted samples vs. 73.0 months in CDKN2A neutral samples, *p* = 0.01) (Figure 4B). Among pathways related to these genes, only MYC signalling was significantly enriched in the non-responders using Gene Set Enrichment Analysis (GSEA) (FDR = 0.04, Figure 4C and Appendix A). Although we observed a significantly elevated expression of MYC in non-responders, MYC amplification was not significantly associated with overall survival (median survival 30.7 months in MYC amplified samples vs. 73.0 months in MYC neutral samples, *p* = 0.068). Overall, we found no significant influence of these copy number changes on overall survival in responders or non-responders alone, as our study was underpowered for these comparisons (Appendix A). However, in responders CDK6 amplification was associated with shorter overall survival (median survival 35.4 months in CDK6 amplified samples vs. 78.5 months in CDK6 neutral samples, *p* = 0.0011).

#### 3.2.3. Mutated Driver Genes Differ Between Responders and Non-Responders

Having established the potential importance of chromosomal level structural variation and gene level copy number variation in response to NAC in EAC, we next moved to assess SNVs, starting with known EAC driver genes (Figure 5A). WGS data showed that 96.2% of responders and 94.3% of non-responders carried at least one non-synonymous somatic mutation in these EAC driver genes. As expected, TP53 (65%), CDKN2A (16%), SMAD4 (15%) and ARID1A (8%) were highly mutated in this cohort [25]. NAV3 was exclusively mutated in non-responders (8/36, 22%, Fisher’s exact test, *p* = 0.01) (Figure 5B). We used the Ensembl Variant Effect Predictor (https://www.ensembl.org/Tools/VEP, release 75; date accessed: 1 November 2020) [52] to predict mutational consequences and found that several mutations, including NAV3 p.V142G and p.D2366N, are likely to be functionally deleterious (Appendix A). We also found mutations in KCNQ3 (4/36 patients, 11%), LRRK2 (4/36 patients, 11%), KRAS (3/36 patients, 8%) and PBRM1 (2/36 patients, 6%) that were unique to non-responders, but these were not statistically significant.

We then determined which mutations might be pathogenic by cross-referencing them with the database of curated mutations [42]. We identified 33 and 54 curated pathogenic mutations in 74% of responders and 75% of non-responders (Appendix A) including three exonic mutations in KRAS (p.G12C, p.G12D and p.G13D). In non-responders KRAS was significantly co-mutated with SMAD4 (Fisher’s exact test, two-sided, *p* = 0.006). Evidence from pancreatic cancer suggests that expression of oncogenic KRAS and loss of SMAD4 cooperate to induce the expression of EGFR and to promote invasion [53].

#### 3.2.4. Potentially Targetable Alterations in Non-Responders

To make sense of the complex genomic aberrations observed in this study and in EAC in general, we combined recurrent CNVs and non-synonymous mutations with the aim of identifying unique genomic aberrations in responders or non-responders. Overall, the mean number of EAC drivers carrying any alteration (SNVs, CNVs and Indels) was higher in non-responders (6.4/patient vs 4.4/patient, Wilcoxon rank-sum test, *p* = 0.007, Figure 5C). We observed that the majority of differentially altered genes were found in non-responders (Figure 5A). Many of these genomic lesions are potentially targetable, and to investigate this further we focused on somatically altered cancer genes which are directly linked to a clinical action in the TARGET database (https://software.broadinstitute.org/cancer/cga/target; date accessed: 1 December 2020.) (Appendix A). Non-responders displayed exclusive focal alterations of genes in this list (Figure 5D), including: amplification of *AURKA* (16/36 non-responders, 44%); *GNAS* (20/36 non-responders, 56%) and *RARA* (12/36 non-responders, 33%); and deletion of *ERFFI1* (12/36 non-responders, 33%), in addition to the previously identified EAC drivers *CDKN2A*, *CCND1*, *EGFR* and *KRAS* (Appendix A). However, we observed chromosome arm amplifications of 20q, containing GNAS and AURKA, in responders. We also found potentially targetable genes exclusively amplified at the focal level in responders: *CEBPA* (13/36 responders, 37%) and *AKT3* (13/36 responders, 37%) were amplified in addition to EAC drivers *ERBB2* and *CCNE1* (Figure 5D). These findings are in contrast to chromosome arm deletions at 19q, containing CCNE1 and CEBPA, which were found in non-responders. We examined the levels of evidence for biomarker-drug associations for our targets using the OncoKB database (https://www.oncokb.org/; date accessed: 1 January 2021.) [41]. *ERBB2, EGFR* and *KRAS* have FDA-approved drugs for use in cancer therapy, whereas *CDKN2A* has biological evidence for targetability, but associated drugs are not yet standard-of-care.

## 4. Discussion

In this study we analyzed whole-genome sequencing data from endoscopic biopsies prior to neoadjuvant chemotherapy in EAC and compared the genomes of responders to non-responders to identify potential genomic determinants of response. We comprehensively profiled CNVs, SNVs and mutational signatures in a cohort powered to identify differences between responders and non-responders. We detected distinct mutational characteristics of EAC between responders and non-responders across the spectrum from large-scale chromosomal alterations to point mutations. Our work characterizes pre-existing genomic alterations that have potential as biomarkers for resistance or sensitivity to NAC.

We found that responders have higher mutational burden, in agreement with a previously published study [26]. Using a neoantigen prediction pipeline, we predicted that an increased mutational burden could lead to more abundant neoantigen recognition in responders. This could serve to bolster anti-tumor immunity as observed in the mutagenic subset of EACs reported previously [24]. Unfortunately, this study was not powered to resolve differences in NAC response between mutational signature subtypes. Although we reliably identified these mutational subtypes, there was not a clear distinction in this cohort. Consistently, we found that non-responders had impaired DNA damage response pathways and had more frequent driver gene mutations and genomic instability, despite having a lower mutation burden. The presence of an immune response related to DNA damage is known to improve survival outcomes and might contribute to this effect, as neoadjuvant therapies are genotoxic and known to stimulate anti-tumor immunity [54].

Our data suggest that responders are dominated by point mutations, whereas non-responders display more copy number changes. Non-responders displayed a unique pattern of copy number changes characterized by chromosome arm deletions and an increased burden of copy number altered segments. This is consistent with an analysis of mutational landscapes in a pan-cancer dataset (not including EAC), which suggests that tumors are dominated by either mutations or copy number changes, but never both [55]. The extremes of this spectrum are occupied by genomically unstable tumors, such as those observed in our cohort of non-responders. Genomic instability has been linked to a suppressed anti-tumor immunity during immunotherapy in gastric cancer [56], whereas in non-small cell lung cancer and melanoma a higher mutation burden is linked to greater neoantigen burden [57] and an improved clinical response to immunotherapy [58,59,60]. Taken together, this suggests that responders may be more likely to benefit from immunotherapy than non-responders and warrants further investigation.

The unique patterns of copy number changes in driver genes have important implications for treatment of chemoresistant EAC patients. These unique amplifications and deletions included potentially druggable signaling axes in EAC, and we found that MYC signaling, RTK-RAS and cell cycle pathways were preferentially mutated in non-responders. This has implications for future clinical trials, as profiling driver mutations prior to neoadjuvant treatment, such as *EGFR*, *CDKN2A* and *CCND1* copy number changes, would be an effective strategy to aid clinical decision-making. This would aid in the identification of patients unlikely to respond to NAC, allowing alternative or concurrent targeted therapies to be considered to exploit these pathway alterations.

Among altered pathways, we highlight cell cycle regulation as a vulnerability in non-responders. G1/S-phase checkpoint genes were disrupted, with *CCND1* and *CDK6* amplification as well as deletion of the *CDK4/6* inhibitors *CDKN2A* and *CDKN2B*. Abnormal expression of CDKs and their partner cyclins is widely reported in esophageal cancer [61,62,63], and *CCND1* amplification and nuclear expression have been shown to correlate negatively with survival [64,65]. Abnormal activity of the CDK/cyclin complexes in esophageal adenocarcinoma has been shown to be a marker of acquired chemo-radio-resistance [65,66]. *CDK4/6* inhibitors could be promising therapeutics for non-responders to NAC with copy number changes in this axis. In particular, *CDK4/6* inhibitors palbociclib, ribociclib and abemaciclib have shown efficacy in in vitro models of EAC [25,67] and promising results in breast cancer, non-small cell lung cancer and melanoma patients [68]. Similarly, the use of ABT-348, a multitarget Aurora kinase and *VEGFR* inhibitor [69], is currently being explored in phase I and II clinical trials in patients with *CDKN2A*-deficient tumors [70,71], suggesting additional targeted therapies to this axis are closer to clinical adoption.

Previous genomic analyses suggest that copy number changes to RTKs are pervasive in EAC, with the potential for targeting with RTK inhibitors specific to the activated pathways, such as trastuzumab and ABT-806 for *ERBB2* (amplified in responders) and *EGFR* (amplified in non-responders), respectively [24,72]. *ERBB2* overexpressing tumors are already treated in the metastatic setting with trastuzumab [73] and recent phase II trials in the perioperative setting are encouraging [74,75]. We found that *EGFR* is uniquely amplified in non-responders, suggesting that anti-*EGFR* antibodies cetuximab or ABT-806 may be useful therapies in these patients. *EGFR*-amplified EAC patients have been shown to particularly benefit from cetuximab [76]. It is well tolerated by EAC patients [77] and has gone through phase III trials as a neoadjuvant therapy in addition to chemotherapy or chemoradiation, showing a modest improvement in recurrence-free survival [78]. However, several other phase III trials have failed to show benefits of cetuximab use in unselected populations of esophageal cancer patients [79,80]. *KRAS* mutations are known to confer resistance to cetuximab in colorectal cancer [81,82], and while this is unclear in EAC due to the rarity of *KRAS* mutations and unselected patient populations [78,83], *KRAS*-mutant tumors in our dataset bore the same mutations in codons 12 and 13 as the resistant colorectal tumors and were non-responders. Taken together these trials underscore the need for careful selection of patient populations to be treated with RTK inhibitors based on mutation status.

Finally, we report that Neuron Navigator-3 (*NAV3*), a known tumor suppressor downstream of *EGFR* [84], is mutated exclusively in non-responders. *NAV3* is a microtubule-binding protein whose expression is regulated by *TP73* and induced by *EGF* in breast cancer cells [84]. In our cohort we observed that *NAV3* was co-mutated with *CDKN2A,* with most mutations being missense. Predictions of neoantigen recognition in lung adenocarcinoma suggest that *NAV3* is one of the most commonly mutated genes with predicted neoantigen recognition in this disease as well [85]. The functional consequences of these mutations in EAC are unclear but we predict several to be functionally deleterious.

*NAV3* is primarily implicated as a metastasis suppressor in multiple cancer types. *NAV3* is upregulated in response to DNA damage in colon carcinoma cells and is involved in the suppression of migration and invasion in vitro [86]. Loss of heterozygosity occurs in colorectal cancer and this associates with lymph node metastasis [87]. *NAV3* expression is attenuated in metastatic colon cancer [86], breast cancer and lung cancer [84] and its knockdown promotes invasive behaviors [84,86], platinum drug resistance [88] and epithelial mesenchymal transition in vitro [86] and enhances metastasis in vivo [84]. The inhibitory effect of *NAV3* on invasion and metastasis may be due to its promotion of slower, directional cell migration as opposed to the random migration observed in *NAV3* knockdown cells, which enhances their ability to explore their environment [84]. Silencing of *NAV3* in vitro also leads to upregulation of *IL-23R* in colorectal [87] and glioma cell lines [89], linked to proinflammatory JAK-STAT signaling. A sizeable proportion of EAC non-responders (22%) carry mutations in *NAV3*, and its status as a unique genetic lesion to this group suggests that *NAV3* mutation could be used as a biomarker to identify some of the patients who fail to respond to NAC. This warrants validation in a larger cohort of patients, including further study of *NAV3* expression and the functional consequences of *NAV3* mutation in EAC.

This study is not without shortcomings. With 63 patients we were able to resolve genomic differences between responders and non-responders at the copy number and mutational level, but had insufficient sample size to fully study the impact of mutational signatures on NAC response. As EACs accrue many genetic alterations and very few are recurrent [25], we lack the power to resolve the significance of rarer mutations on survival and to determine rarer co-mutations. However, even with limited sample size, WGS was able to identify KRAS mutations in non-responders, which are frequently associated with treatment resistance in colorectal cancer [81,82], demonstrating the robustness of our approach. Despite these shortcomings, our dataset of responders and non-responders to NAC is the largest of its kind in EAC and represents a step forward in our understanding of the genetic determinants of NAC resistance.

## 5. Conclusions

In summary, we identified genetic features and mutations that are uniquely associated with response to NAC. This indicates the presence of a subset of patients harboring pre-existing mutations that confer resistance to NAC. Importantly, these mutations are potentially clinically actionable, with a variety of drugs in clinical trials to support a targeted therapy strategy—an approach that has previously met with success in metastatic EAC patients [72]. We envision a treatment pipeline that incorporates driver mutation profiling in EAC, combining response prediction with targeted therapies to enhance response to NAC and improve survival outcomes.

## Figures and Tables

**Figure 1 cancers-13-03394-f001:**
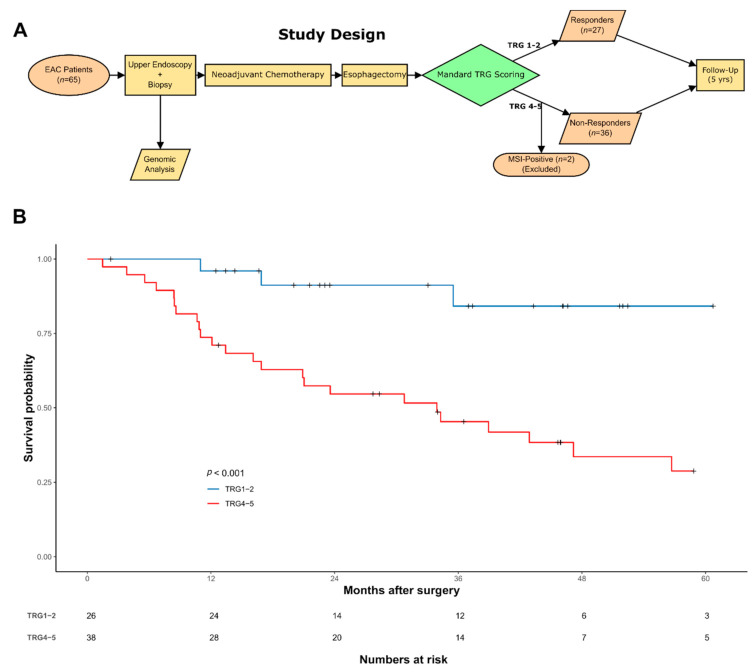
Outline of the cohort and analyses performed. (**A**) Description of the study design. (**B**) Kaplan–Meier of overall survival (*n* = 64) for responders (blue line) and non-responders (red line). Number of cases at risk are detailed in the table.

**Figure 2 cancers-13-03394-f002:**
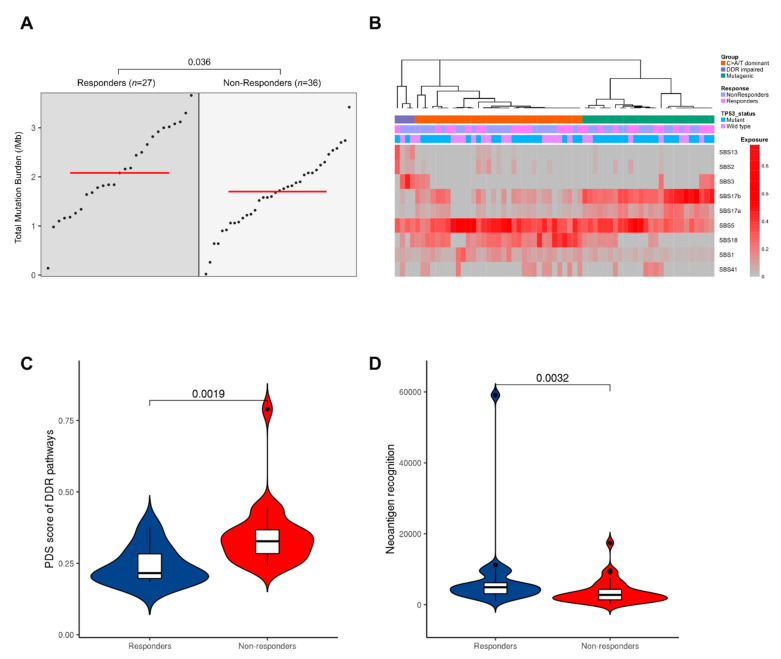
Mutational landscape of NAC response. Responders have a higher mutation burden and neoantigen recognition potential. (**A**) Group dot plot of mutation per megabase. Each dot represents a patient with the red line marking the group median. (**B**) Clustering of the nine mutational signatures in our patient samples as previously described by Secrier et al. [24]. (**C**) Pathway deregulation scores (PDS) calculated using gene expression values (log2 normalized) of available RNA-seq samples for DDR pathways. (**D**) Neoantigen recognition potential scores. Only neoantigens with recognition potential above 1 are shown.

**Figure 3 cancers-13-03394-f003:**
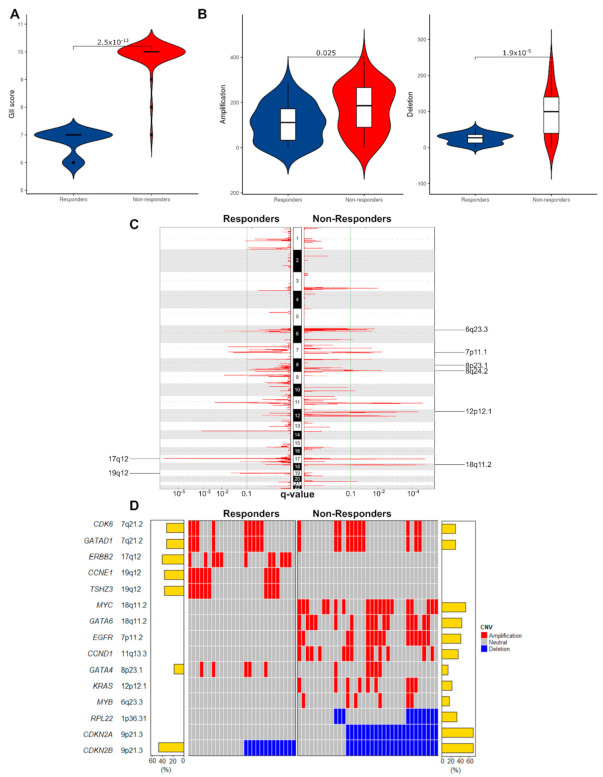
Non-responders have less stable genomes and unique patterns of copy number change in EAC driver genes. (**A**) Proportions of the genome affected by copy-number changes (Genomic Instability Index, GII). Non-responders showed a higher level of genomic instability (*p* = 2.5 × 10^−13^). (**B**) Violin plots depicting the frequency of all amplifications and deletions in responders and non-responders. (**C**) Amplified peak regions across the genome plotted for responders vs non-responders (*n* = 63) using GISTIC2.0 (FDR < 0.1). Amplifications unique to each group are labeled. (**D**) Oncoplot of recurrently amplified/deleted EAC drivers among responders and non-responders identified by GISTIC2.0 (FDR < 0.1).

**Figure 4 cancers-13-03394-f004:**
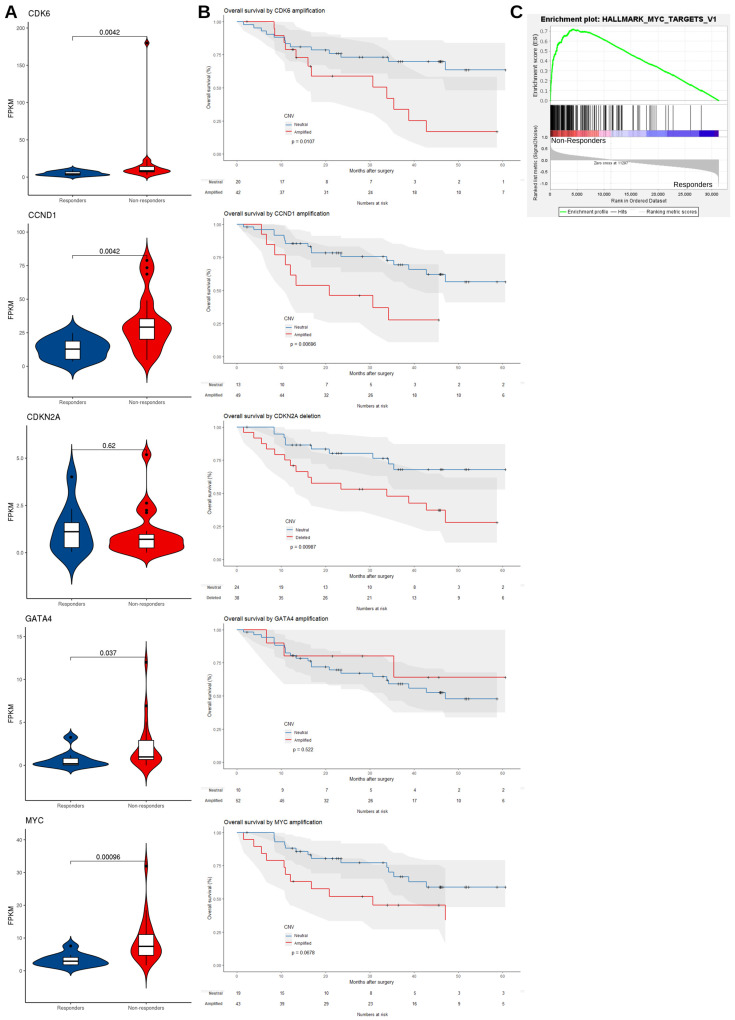
Amplified EAC driver genes are overexpressed in non-responders and copy number alterations associate with poor survival. (**A**) Violin plots comparing mRNA expression levels (FPKM) in matched RNA-Seq data (*n* = 9 responders, *n* = 21 non-responders) for copy number altered EAC driver genes in responders and non-responders. *p*-values were based on one-tailed Wilcoxon rank-sum test. (**B**) Kaplan–Meier plot comparing overall survival of patients with *CDK6*, *CCND1, CDKN2A, GATA4* and *MYC* copy number changes (red) vs. neutral patients (blue). (**C**) Gene Set Enrichment Analysis (GSEA) of *MYC* target genes in available RNA-Seq data (*n* = 30).

**Figure 5 cancers-13-03394-f005:**
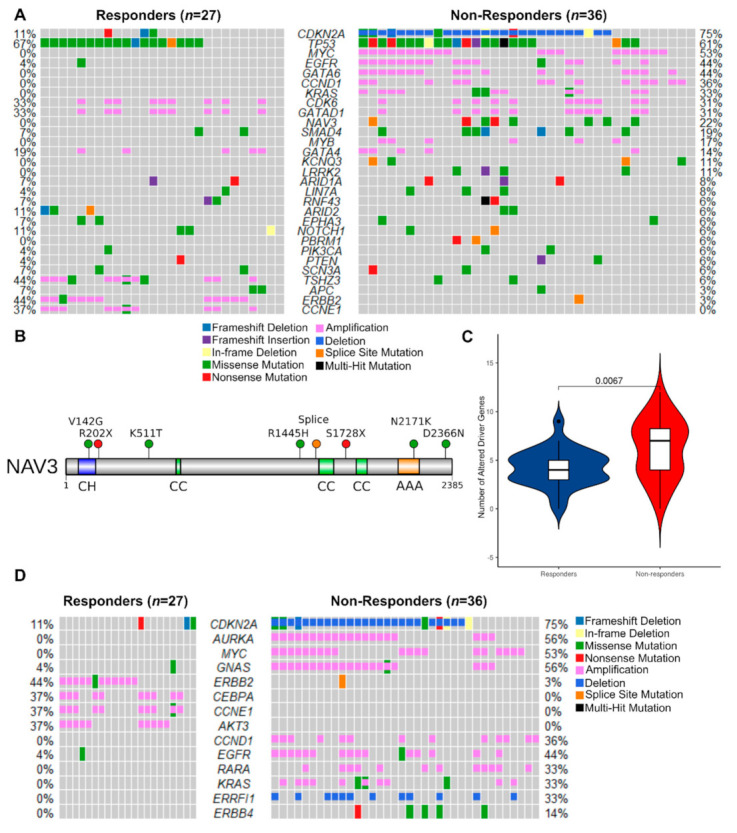
Non-responders have more EAC driver alterations of all types, including exclusive mutation of the tumor suppressor *NAV3*. (**A**) Oncoplot of SNVs, indels and CNVs combined in responders vs. non-responders (*n* = 63). Genes shown are the subset of the 76 EAC driver genes described in Frankell et al. [25] that were mutated in at least 5% of either group. Percentages of responders or non-responders with driver gene mutations are shown next to the corresponding row. (**B**) Protein-level diagram of mutations in the coding sequence of *NAV3*, which was exclusively mutated in non-responders. Domains are labeled as follows: CH—calponin homology; CC—coiled coil; AAA—ATPase associated with diverse cellular activities. Mutational sites are shown as lollipops color-coded according to the type of mutation. (**C**) Violin plots comparing frequency of all alterations (SNVs, indels and CNVs) in EAC driver genes per sample in responders vs. non-responders. (**D**) Oncoplot of SNVs, indels and CNVs combined in TARGET database genes, which are associated with a clinical action in cancer (*n* = 63). Genes that were mutated in at least 5% of either group are shown.

**Table 1 cancers-13-03394-t001:** Clinicopathological data for the study cohort according to response to NAC.

Variable	Category	Non-Responders (*n* = 38)	Responders (*n* = 27)	Overall (*n* = 65)	*p*-Value (X^2^ test)
Age		66.25 (15.3)	64.30 (12.2)	65.00 (14)	0.739
Gender	Female	5 (13.2)	2 (7.4)	7 (10.8)	0.741
	Male	33 (86.8)	25 (92.6)	58 (89.2)	
cT Stage	T1	0 (0.0)	1 (3.7)	1 (1.5)	0.485
	T2	5 (13.2)	5 (18.5)	10 (15.4)	
	T3	31 (81.6)	19 (70.4)	50 (76.9)	
	T4	2 (5.3)	1 (3.7)	3 (4.6)	
	Missing	0 (0.0)	1 (3.7)	1 (1.5)	
cN Stage	N0	8 (21.1)	7 (25.9)	15 (23.1)	0.494
	N1	24 (63.2)	12 (44.4)	36 (55.4)	
	N2	5 (13.2)	6 (22.2)	11 (16.9)	
	N3	1 (2.6)	1 (3.7)	2 (3.1)	
	Missing	0 (0.0)	1 (3.7)	1 (1.5)	
Tumor Location	GOJ	19 (50.0)	16 (59.3)	35 (53.8)	0.627
	Esophagus	19 (50.0)	11 (40.7)	30 (46.2)	
ypT Stage	T0	1 (2.6)	17 (63.0)	18 (27.7)	<0.001 *
	T1	3 (7.9)	5 (18.5)	8 (12.3)	
	T2	4 (10.5)	1 (3.7)	5 (7.7)	
	T3	24 (63.2)	4 (14.8)	28 (43.1)	
	T4	6 (15.8)	0 (0.0)	6 (9.2)	
ypN Stage	N0	9 (23.7)	17 (63.0)	26 (40.0)	<0.001 *
	N1	6 (15.8)	4 (14.8)	10 (15.4)	
	N2	14 (36.8)	2 (7.4)	16 (24.6)	
	N3	8 (21.1)	0 (0.0)	8 (12.3)	
	Missing	1 (2.6)	4 (14.8)	5 (7.7)	
ypM Stage	M0	35 (92.1)	27 (100.0)	62 (95.4)	0.371
	M1	3 (7.9)	0 (0.0)	3 (4.6)	
Treatment Regimen	CarboTaxol	2 (5.3)	0 (0.0)	2 (3.1)	<0.001 *
	CF	1 (2.6)	1 (3.7)	2 (3.1)	
	CX	0 (0.0)	4 (14.8)	4 (6.2)	
	CROSS	0 (0.0)	2 (7.4)	2 (3.1)	
	ECarboX	0 (0.0)	1 (3.7)	1 (1.5)	
	ECF	0 (0.0)	1 (3.7)	1 (1.5)	
	ECOx	1 (2.6)	0 (0.0)	1 (1.5)	
	ECX	29 (76.3)	11 (41.0)	39 (60.0)	
	ECX + Bevacizumab	2 (5.3)	0 (0.0)	2 (3.1)	
	EOX	1 (2.6)	3 (11.1)	4 (6.2)	
	LEO	1 (2.6)	0 (0.0)	1 (1.5)	
	CAPOX	0 (0.0)	1 (3.7)	1 (1.5)	
	Missing	3 (7.9)	1 (3.7)	4 (6.2)	

Data presented as absolute number (%) and median (IQR), * *p* < 0.05 indicates a Mann–Whitney U test *p*-value. Treatment regimens: CarboTaxol—Carboplatin + Paclitaxel; CF-Cisplatin + 5-Fluorouracil; CX-Cisplatin + Capecitabine; CROSS-Carboplatin + Paclitaxel with concurrent radiotherapy 41.4Gy; ECarboX-Epirubicin + Carboplatin + Capecitabine; ECF-Epirubicin + Cisplatin + 5-fluorouracil; ECOx-Epiru-bicin + Cisplatin + Oxaliplatin; ECX-Epirubicin + Cisplatin + Capecitabine; EOX-Epirubicin + Oxaliplatin + Capecitabine; LEO-Lapa-tinib + Epirubicin + Oxaliplatin; CAPOX-Oxaliplatin + Capecitabine.

## Data Availability

The WGS and RNA-seq data generated in this study are available from the European Genome-Phenome Archive (https://ega-archive.org/ accessed on 10 June 2021) under accession number EGAD00001007493.

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
