# Peer review of "Genomic Analysis of Response to Neoadjuvant Chemotherapy in Esophageal Adenocarcinoma"

_cancers, 2021, doi:10.3390/cancers13143394_

Round 1

Reviewer 1 Report

  • The statement that NAC provides a 5% survival benefit at 5 years is an underestimation. Already the meta-analysis by Ronellenfitsch et al. (2013), which you cite, showed a benefit of about 9% at 5 years. This magnitude of overall survival benefit was confirmed by more recent trials with intensified regimens (FLOT-4 trial: 9% at 5 years, CROSS trial: 10% at 10 years in the adenocarcinoma subgroup). Likewise, the proportion of patients experiencing "a very good response" is underestimated. For example, the proportion of patients with Becker TRG 1a/b, which roughly corresponds to Mandard gardes 1/2, is 37% in the experimental arm of the phase 2 part of the FLOT-4 trial (Al-Batran et al., Lancet Oncol 2016).  Please correct these statements throughout the manuscript and cite the studies where appropriate.
  • You should explain the Mandard TRG system and provide the definitions for the single grades. Otherwise, it is difficult to put your results into the given context.
  • Please provide details regarding the neoadjuvant therapies the patients in your study received: Chemotherapy or chemoradiotherapy? Which regimens and which dose intensity?
  • The rationale of excluding MSI high tumors from your analysis is unclear to me. Although you are right that they are potential candidates for immunotherapy, MSI testing is not routinely done at present, and thus these patients would probably receive "standard" neoadjuvant chemotherapy as of now. Therefore, their respective genetic alterations determining response and outcome would still be interesting.
  • The results section is rather long and would benefit from shortening of some degree.
  • You point out that ERBB2 (Her2neu) and EGFR could be good therapeutic targets, given that therere are approved drugs with proven efficacy in other entities. However, numerous trials have not been able to show a clear therapeutical benefit for EGFR targetting with Cetuximab (compare: Ilson DH. Is there a future for EGFR targeted agents in esophageal cancer? Ann Oncol. 2018 Jun 1;29(6):1343-1344. doi: 10.1093/annonc/mdy135. PMID: 29668836.), so that it must be assumed that the drug has no further indication in esophageal adenocarcinoma. On the other hand side, Her2neu targetting with trastuzumab is a proven therapeutic option for Her2neu overexpressing tumors. It is standard of care in the metastatic setting (Bang YJ, Van Cutsem E, Feyereislova A, Chung HC, Shen L, Sawaki A, Lordick F, Ohtsu A, Omuro Y, Satoh T, Aprile G, Kulikov E, Hill J, Lehle M, Rüschoff J, Kang YK; ToGA Trial Investigators. Trastuzumab in combination with chemotherapy versus chemotherapy alone for treatment of HER2-positive advanced gastric or gastro-oesophageal junction cancer (ToGA): a phase 3, open-label, randomised controlled trial. Lancet. 2010 Aug 28;376(9742):687-97. doi: 10.1016/S0140-6736(10)61121-X.) and being evaluated in the perioperative setting (Rivera F, Izquierdo-Manuel M, García-Alfonso P, Martínez de Castro E, Gallego J, Limón ML, Alsina M, López L, Galán M, Falcó E, Manzano JL, González E, Muñoz-Unceta N, López C, Aranda E, Fernández E, Jorge M, Jiménez-Fonseca P. Perioperative trastuzumab, capecitabine and oxaliplatin in patients with HER2-positive resectable gastric or gastro-oesophageal junction adenocarcinoma: NEOHX phase II trial. Eur J Cancer. 2021 Mar;145:158-167. doi: 10.1016/j.ejca.2020.12.005.; Hofheinz RD, Hegewisch-Becker S, Kunzmann V, Thuss-Patience P, Fuchs M, Homann N, Graeven U, Schulte N, Merx K, Pohl M, Held S, Keller R, Tannapfel A, Al-Batran SE. Trastuzumab in combination with 5-fluorouracil, leucovorin, oxaliplatin and docetaxel as perioperative treatment for patients with human epidermal growth factor receptor 2-positive locally advanced esophagogastric adenocarcinoma: A phase II trial of the Arbeitsgemeinschaft Internistische Onkologie Gastric Cancer Study Group. Int J Cancer. 2021 May 21. doi: 10.1002/ijc.33696. These aspects need to be discussed.

Reviewer 2 Report

Review of the Manuscript “Genomic Analysis of Response to Neoadjuvant Chemotherapy in Esophageal Adenocarcinoma”

This compelling analysis is a pre-treatment genome sequencing of patients planned for multimodal therapy for esophageal adenocarcinoma. The authors intended to identify genetic characteristics that are particularly associated with response to neoadjuvant chemotherapy.

Response to neoadjuvant therapy is still challenging to predict, and the mechanism of non-responding is not yet completely understood. Unlike other tumor entities, evidence regarding genetic mechanisms is scarce. Thus, this scientific work might be of great interest to the readers of "Cancers". This paper is of excellent writing style and enjoyable to read with nice figures and tables. I commend the authors for their effort.

I suggest minor revision before publication:

-        How did you select the 65 cases from the OCCAMS dataset?

-        In the discussion, you mentioned a underpower for an analysis of mutational signature subtypes. Did you perform a power analysis?

-        What Chemotherapy regimens did the patients receive?

-        In Table 1, at variable “age” IQR is missing.

Round 2

Reviewer 1 Report

Most of the changes I suggested have been carried out. Therefore, I can recommend publication of the paper.